# Neuroscience and Neuroimmunology Solutions for Osteoarthritis Pain: Biological Drugs, Growth Factors, Peptides and Monoclonal Antibodies Targeting Peripheral Nerves

**Ali Mobasheri** [1,2,3]

1 Research Unit of Medical Imaging, Physics and Technology, Faculty of Medicine, University of Oulu, FI-90014 Oulu, Finland; ali.mobasheri@oulu.fi or ali.mobasheri@imcentras.lt or a.mobasheri@umcutrecht.nl
2 Department of Regenerative Medicine, State Research Institute Centre for Innovative Medicine, LT-08406 Vilnius, Lithuania
3 Departments of Orthopedics, Rheumatology and Clinical Immunology, University Medical Center Utrecht, 3508 GA Utrecht, The Netherlands

**Abstract:** Neuroscience is a vast discipline that deals with the anatomy, biochemistry, molecular biology, physiology and pathophysiology of central and peripheral nerves. Advances made through basic, translational, and clinical research in the field of neuroscience have great potential for long-lasting and beneficial impacts on human and animal health. The emerging field of biological therapy is intersecting with the disciplines of neuroscience, orthopaedics and rheumatology, creating new horizons for interdisciplinary and applied research. Biological drugs, growth factors, therapeutic peptides and monoclonal antibodies are being developed and tested for the treatment of painful arthritic and rheumatic diseases. This concise communication focuses on the solutions provided by the fields of neuroscience and neuroimmunology for real-world clinical problems in the field of orthopaedics and rheumatology, focusing on synovial joint pain and the emerging biological treatments that specifically target pathways implicated in osteoarthritis pain in peripheral nerves.

**Keywords:** neuroscience; rheumatology; osteoarthritis (OA); pain; peripheral nerve; biological drug; growth factor; therapeutic peptide; monoclonal antibody (mAb); ion channel



## 1. Introduction

Neuroscience is a branch of biology and medicine that deals with the anatomy, physiology, biochemistry and molecular biology of the central and peripheral nervous systems. It deals with the structure and function of central and peripheral nerves and their relationship with other cells, tissues and organs that are associated with nerves. For many decades the field of neuroscience was viewed as a separate discipline, isolated and independent of other areas of medicine. However, in recent years, the fields of neuroscience, oncology, immunology and metabolism have intersected, creating exciting opportunities for multi-disciplinary collaborative research in basic, translational and clinical sciences. For example, the integration of cancer-related neuroscience research is opening up new horizons in cancer-related neurotoxicity to investigate disease mechanisms, develop new drugs and implement novel pharmacological interventions. The integration of neuroscience in the fields of immunology and the emerging field of immunometabolism is helping to build a clearer picture of the role of the nervous system in immune function and inflammation [1]. The role of the peripheral and central nervous systems in tumor growth and metastasis is also increasingly well recognized and it is becoming widely accepted that many neoplastic and auto-immune diseases have an important neuroscience component [2,3]. Neuroscience is impacting other areas of medicine, including immunology, endocrinology, metabolism, orthopedics and musculoskeletal sciences [4].

The nervous system is now recognized to be a component of many age-related and inflammatory diseases. The rapidly expanding aging population [5] and changing de-

mographics [6,7] highlight opportunities for studying the role of the nervous system in different diseases such as cancer, immunological disorders, inflammatory conditions, metabolic and musculoskeletal diseases. This communication focuses on the solutions that neuroscience and neuroimmunology are able to offer for problems in the fields of orthopaedics and rheumatology, and specifically addresses the issue of arthritis pain. This paper will summarize emerging treatments that specifically target nociceptive pathways implicated in arthritis pain in peripheral nerves. Many of the emerging treatments for osteoarthritis (OA) are biological and intra-articular. Neuroscience has already offered new treatment targets for OA by unraveling the crucial role of nerve growth factor (NGF) in the development and evolution of pain. More research is needed to understand the roles of the central and peripheral nervous systems in OA pain and identify safer and more effective pain medications.

## 2. Osteoarthritis (OA)

OA is the most prevalent form of arthritis globally and is the leading cause of physical disability and the primary source of health and social societal cost in older adults [8]. OA is a serious disease [9]. According to the World Health Organization (WHO), OA affects millions of people worldwide. Recent estimates suggest that OA affects at least 7% of the global population, which represents more than 500 million people worldwide, with women disproportionately affected by the condition [10], especially after menopause [11–13]. Although OA is primarily related to aging [14], it is also associated with a wide variety of modifiable and non-modifiable risk factors that include: overweight and obesity [15,16], sedentary behavior [17] and lack of physical exercise [18]. OA is a syndrome with a multifactorial etiology [19]. In addition to the primary risk factors of aging, obesity, the female gender, and genetics, other inciting risk factors for OA may include previous joint trauma or history of repetitive joint injuries or even the presence of metabolic syndrome and endocrine disease [20]. However, OA is primarily a biomechanical disease [21]. However, in addition to biomechanical factors [22], there are inflammatory [23] and metabolic [24] factors that play dominant roles in the initiation and progression of OA.

Many patients with osteoarthritis exhibit comorbidities such as obesity, low-grade systemic inflammation, diabetes mellitus and depression [25–28]. These comorbidities can significantly influence the course of osteoarthritis, and the intensity and frequency of joint pain, which is thought to be influenced by depression [29–31]. Research has only recently begun to focus on the significance of such factors in OA pain. We know that changes in peripheral joint innervation are partly responsible for degenerative alterations in joint tissues which contribute to the development of OA [32,33]. This is why targeting NGF and peripheral innervation in painful arthritic joints with targeted biological drugs is timely and important [34–36]. Ion channels and G-protein coupled receptors (which are often ion channels themselves) in peripheral innervation of painful osteoarthritis joints must therefore be a high priority target for therapeutic monoclonal antibodies (mAbs) and biological drugs that are administered intra-articularly rather than injected subcutaneously or systemically [37–40].

Another promising area for future development focuses on nanobodies and bispecific antibodies. Nanobodies, originally identified in camelids, are a class of therapeutic proteins based on single-domain antibody fragments that contain the unique structural and functional properties of a naturally-occurring heavy chain-only antibodies [41–44]. Bispecific antibodies on the other hand possess binding specificity for two different target molecules and have recently been developed for targets in a range of autoimmune diseases, such as RA, systemic lupus erythematosus, and psoriasis, and tested in clinical trials [45].

## 3. Existing Recommended Treatments for OA

Despite the heavy burden of OA on individuals, families, and healthcare systems, there are currently no disease-modifying OA-specific treatments authorized for clinical use [46]. According to the experts and the treatment guidelines that have been published recently,

Everyone should receive education teaching them to be active, exercise and manage their weight. Some individuals may benefit from anti-inflammatory drugs or interarticular injections and a few need surgery (Figure 1). However, as the aging population expands and the number of people with OA rises across the world, the number of individuals that require surgery and joint replacement is likely to rise. Inevitably, this will place increased pressure on the need for orthopedic surgery and the cost to individuals and payers, including healthcare systems and insurance companies.

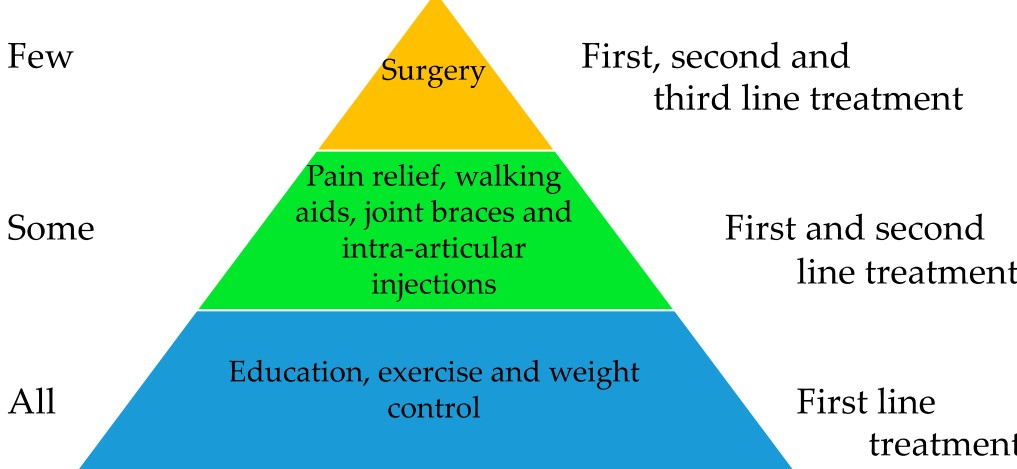

**Figure 1.** The osteoarthritis (OA) treatment pyramid outlining the first, second and third-line treatment for patients with this disease. All patients will benefit from education, exercise and weight control. Some patients that benefit from anti-inflammatory drugs and intra-articular injections. A small number of individuals will require surgery and joint replacement.

Unfortunately, many OA clinical trials have failed and continue to fail at the phase II stage. Most trials conducted to date have produced inconclusive results [47]. This means that the novel pharmacological agents, disease-modifying OA drugs (DMOADs) [48–50] and biological interventions that are currently being tested must have an impact on joint structure (i.e., articular cartilage) and the symptoms of pain, even if the trials include surrogate endpoints and post-marketing confirmatory data under the accelerated drug approval regulations set forth by the Food and Drug Administration (FDA) [51].

## 4. Emerging Biological Treatments for OA

Biological treatments that target OA, attenuate catabolic activity and promote articular cartilage repair include purified or recombinant growth factors, including fibroblast growth factor 18 (FGF-18), also known as Sprifermin [52–54], bone morphogenic protein 7 (BMP7) [55,56], transforming growth factor β1 (TGF-β1) [57–59], neuropeptides [60,61], humanized mAbs that target NGF and vascular endothelial growth factor (VEGF) (i.e., tanezumab (Pfizer) and fasinumab (Teva/Regeneron) against NGF and bevacizumab (Genentech/Roche) and ranibizumab (Genentech/Roche) against VEGF) [62–66], gene and cell therapy incorporating allogeneic cells and protein production platforms that overproduce produce growth factors (i.e., Kolon TissueGene's TissueGene-C, overproducing TGF-β1) [67–71] (Figure 2). A detailed discussion of each of these areas and intra-articular therapies is way beyond the scope of this communication and readers are directed to several comprehensive reviews [72–75]. The following section will focus on biological treatments that specifically target ion channels and nociception in peripheral nerves.

**Growth Factors**

| | |
|---|---|
| Recombinant growth factors | FGF-18 (Sprifermin) TGF-β1 |
| Growth factors delivered using cell therapy | TissueGene-C (TGF-β1) |

**Monoclonal Antibodies**

| | |
|---|---|
| TNF-α inhibitors | Etanercept Adalimumab Infliximab |
| IL-1β inhibitors | AMG108 Canakinumab ABT-981 Anakinra Lutikizumab |

**Anti-Inflammatory Cytokines**

IL-4
IL-10
IL-13
IL-1Ra

**Anti-NGF Antibodies**

Tanezumab
Fasinumab

**Figure 2.** Biological intra-articular treatments for OA include growth factors, humanized monoclonal antibodies (mAbs), anti-inflammatory cytokines and cytokine receptor blockers, cell and gene therapy that results in over-expression of growth factors or blocks cytokine receptors or stimulates chondrogenic gene expression. Biological intra-articular therapies also include mAbs that inhibit nerve growth factor (NGF) and interfere with NGF signaling.

Among the many biological mediators involved in OA pain, NGF is one of the most promising because mAbs that block and neutralize NGF significantly reduce OA pain [76–78] (Figure 2). The development of mAbs that inhibit NGF for OA pain would not have been possible without significant input from the fields of neuroscience and immunology. Several studies show that neutralization of IL-1β and TNF-α may reduce OA pain [79,80]. However, the impact of IL-1β inhibition on pain has been shown to be very modest in the Anakinra (Amgen) trial [81] and the AMG 108 (Amgen) trial [82]. The same outcome was observed in subsequent clinical trials of ABT-981—lutikizumab (AbbVie)—anti-IL-1 α/β dual variable domain immunoglobulin conducted by AbbVie in erosive hand OA [83] and knee OA with synovitis [84]. It is important to note that in some of these trials the mAbs were delivered subcutaneously, which is not ideal because it is not

delivered directly to the arthritic joint. This is a very important weakness of some of these clinical trials and has not been discussed and debated widely.

## 5. Ion Channels and Pain

In the last two decades research from the converging fields of nociception and ion channel biology has helped to identify many ion channels involved in nociception and pain. Originally the research focused on the capsaicin receptor and its possible role in thermosensation, ATP-gated channels, proton-gated channels, and nociceptor-specific sodium channels [85–89]. In 2006, humans with congenital insensitivity to pain (CIP) were found to lack functional NaV1.7 channels [90] and new research began to focus on the sodium channel family as mediators of pain [91]. This was followed by a rush to develop selective inhibitors of NaV1.7 channels with the ultimate goal of producing effective analgesics without the problems of addiction and tolerance associated with opioids [92]. However, in the years that have elapsed since then, it has been increasingly clear that translation from in vitro studies and preclinical studies conducted with rodent models to humans is extremely challenging. Many of the excellent pharmacology studies conducted with in vitro and preclinical models do not translate into in vivo analgesic efficacy [93]. Furthermore, many of the drug pipelines do not test promising candidates in an appropriate translational large animal model and jump straight from mice to men [94]. Recent research using various inflammatory models has shown that acute administration of peripherally restricted NaV1.7 inhibitors can produce analgesia but only when administered in combination with an opioid [95]. Despite this drawback, research in this area has opened up new opportunities for screening natural products including spider, scorpion and snake venom for peptides that can be used as inhibitors of sodium channels and other ion channels with analgesic potential.

## 6. Drug Pipelines with Putative Ion Channels Targets

The US-based drug company Flexion Therapeutics has a biological drug targeting the Nav1.7 sodium channel. In April 2020 Flexion announced promising preclinical data to support the development of FX301, a locally administered Nav1.7 inhibitor candidate for post-operative pain. Although this is not an OA drug, its development has opened up exciting new opportunities for targeting ion channels involved in OA pain.

TissueGene-C, developed by the US-based company Kolon TissueGene is another exciting development in cell and gene therapy specifically developed for the treatment of knee OA [69]. In this unique product, transfected and irradiated protein packaging cell lines are used as "cellular factories" for the production of therapeutic TGF-β1. TissueGene-C is a unique combination of cell and gene therapy targeting knee OA. The treatment strategy is simple and is achieved through a single intra-articular injection of joint-derived chondrocytes mixed with irradiated GP2-293 cells, a protein production platform derived from HEK293 cells. The general concept for TissueGene-C is presented in Figure 2. Effectively, the GP2-293 cells in TissueGene-C are a protein-producing tool and "cellular factory". We have previously emphasized that native patient-derived primary chondrocytes do not have the capacity to over-produce growth factors such as TGF-β1 in the high quantities needed for effective cellular therapy and regenerative applications. TissueGene-C has been shown to promote an anti-inflammatory micro-environment in OA via polarization of M2 macrophages leading to pain relief and structural improvement [70]. Although the mechanism of action of TissueGene-C is currently not thought to target ion channels directly, this possibility has not been ruled out. The impact that TissueGene-C has on the inflammatory micro-environment of the joint, specifically the reduction in pain, implies that it has biological impacts across the entire cellular taxonomy of the joint (Figure 3) including neurons in peripheral nerves, which warrants further investigation into its putative action on joint innervation.

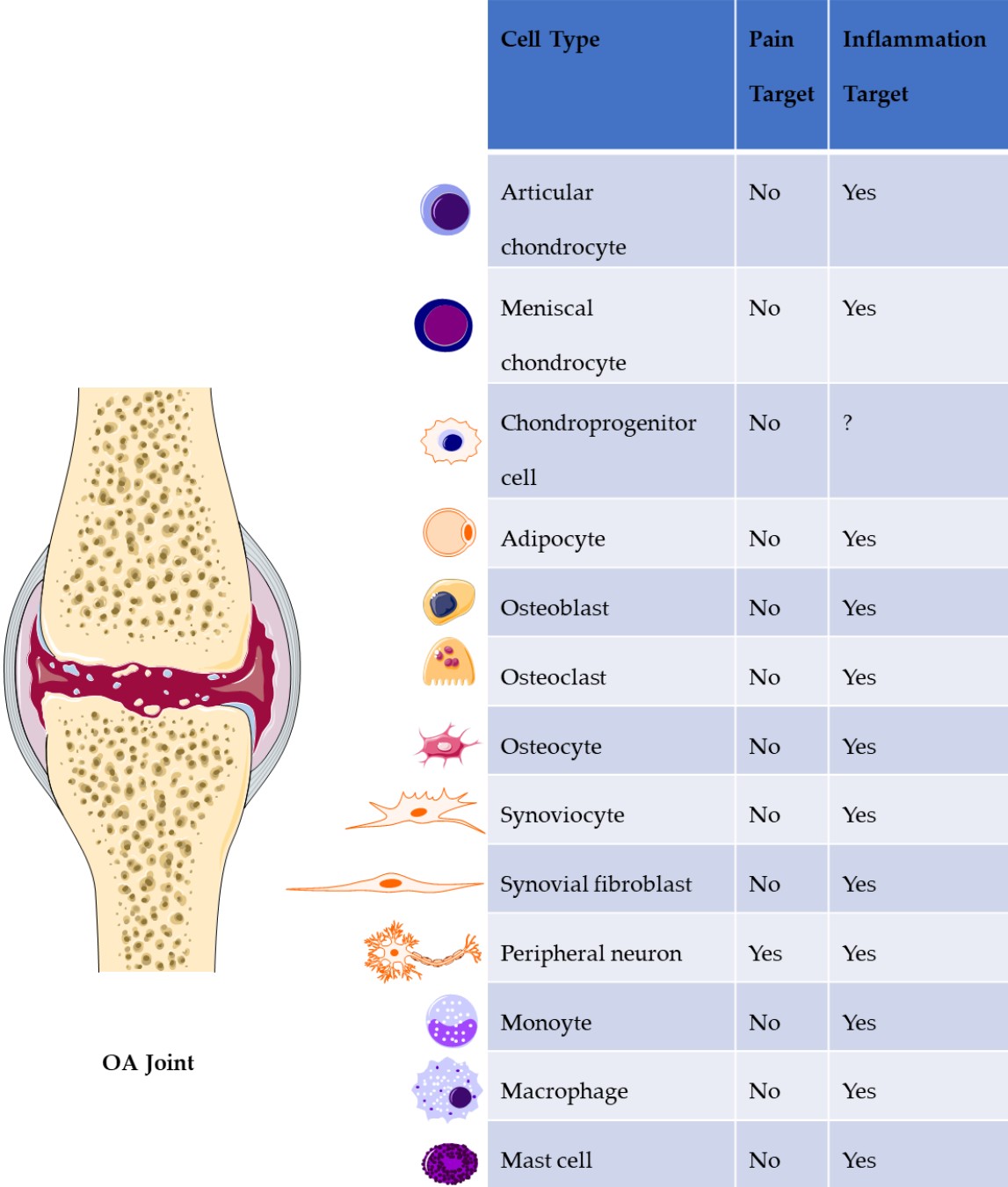

| Cell Type | Pain Target | Inflammation Target |
|---|---|---|
| Articular chondrocyte | No | Yes |
| Meniscal chondrocyte | No | Yes |
| Chondroprogenitor cell | No | ? |
| Adipocyte | No | Yes |
| Osteoblast | No | Yes |
| Osteoclast | No | Yes |
| Osteocyte | No | Yes |
| Synoviocyte | No | Yes |
| Synovial fibroblast | No | Yes |
| Peripheral neuron | Yes | Yes |
| Monoyte | No | Yes |
| Macrophage | No | Yes |
| Mast cell | No | Yes |

**OA Joint**

**Figure 3.** The cellular taxonomy of the synovial joint in OA. Many cell types are involved in the process of OA development and progression but of the research has focused on articular chondrocytes. A more comprehensive cellular taxonomy should include all the cell types that are found within the synovial joint and greater emphasis on the phenotypic and functional plasticity of peripheral neurons. In this scheme peripheral nerves are the primary targets of biological drugs targeting OA pain. However, many of the other cell types produce and secrete pro-inflammatory cytokines such as IL-1β, TNF-α, IL-6, IL-8 and growth factors such as nerve growth factor (NGF) and vascular endothelial growth factor (VEGF).

## 7. mAbs Targeting Ion Channels

The expanding collection of therapeutic mAbs represent a rapidly growing class of biological drugs [96]. Whether mAbs are injected locally or delivered systemically, they are becoming an important arsenal in precision medicine and targeted cancer therapy [97–99]. Many of the recently developed therapeutic mAbs have high specificity and affinity for

their target antigen, which is often present on the cell surface [96]. This is why it is crucially important to learn more about the surfaceome and membranome of chondrocytes and other cell types within the synovial joint, which has been an active area of research in our laboratory for the last few years [100,101]. The establishment of a comprehensive surfaceome and membranome for chondrocytes and other synovial joint cells may combine multi-omics approaches with conventional by chemical, immunological, cellular and in silico techniques [102]. An important sub-component of the surfaceome and membranome is the channelome, which is the complete set of ion channels and porins expressed in a cell [103]. Ion channels are a large family of transmembrane proteins that control ion transport across the cell membrane. They are actively involved in almost all cell biological processes in both health and disease and are widely considered as prospective therapeutic targets in many cardiovascular, respiratory, musculoskeletal and neurological diseases [104–111]. The channelome has been partially been defined in chondrocytes using a combination of proteomic and electrophysiological approaches [112–115]. However, thus far, no antibody-based drug targeting ion channel has been developed for clinical use [116,117]. Despite this limitation, there is huge potential for new research and development in this area. The technologies for designing and producing antibodies have become increasingly streamlined, scalable, simplified, practical and affordable. We need to focus on identifying pain targets and developing mAbs against them. We must also take advantage of the technological advances in protein and antibody engineering, cell and gene therapy, biomaterials and nanotechnology [118–122] and innovate in this area to develop new diagnostics and therapeutics for the benefit of people with OA and other forms of arthritis.

## 8. Discussion

The field of rheumatology received a massive boost from basic research conducted in the field of oncology following the introduction of inhibitors of tumor necrosis factor α (TNF-α) for the treatment of a range of rheumatological conditions, beginning with RA [123–125]. OA is not the same as RA but the two diseases share some important characteristics. At the present time, OA remains highly problematic as a disease entity. Existing drugs only address symptoms of OA and there are no approved DMOADs [126,127]. OA drug development is hampered by the lack of sensitive outcome measures and there are only a handful of biomarkers that can be used to test the efficacy of new drugs [128].There are at present no early biomarkers of early OA [129,130]. Several Phase II OA clinical trials have recently failed but the therapeutic pipeline contains several promising biological candidates and it is hoped that a few symptoms modifying drugs may be approved within the next 3 to 5 years. Many future OA treatments are likely to be biological and developed for intra-articular delivery. However, we will need better biomarkers to assess the efficacy of these treatments and develop intra-articular biological treatments that are targeted to intra-articular phenotypes of OA (i.e., articular cartilage, synovitis and sub-chondral bone phenotypes). At the present time, we do not possess the tools for phenotyping and stratification of heterogeneous OA cohorts. However, as we develop these tools using deep phenotyping and multi-omics-driven approaches, we will be closer to achieving patient stratification for more effective targeting of biological drugs (Figure 4).

We must exploit opportunities to develop novel biological drugs for intra-articular injection and apply them as early as possible. Synovial biopsies and biomarkers have been very useful for guiding rheumatology practice, monitoring disease progression and response to therapy in RA and other rheumatic diseases. The situation is going to be much more challenging in the case of OA. Synovial fluid and synovial biopsies are difficult to collect and require invasive procedures but the collection and banking of these will be needed for biomarker studies and future OA drug development. We also need to develop more sensitive methods to non-invasively assess synovial joint inflammation using magnetic resonance imaging and ultrasound. In conclusion, developing DMOADs remains challenging and this is an area that will benefit from interdisciplinary collaboration to develop new ideas into therapeutic innovations and treatments (Figure 5).

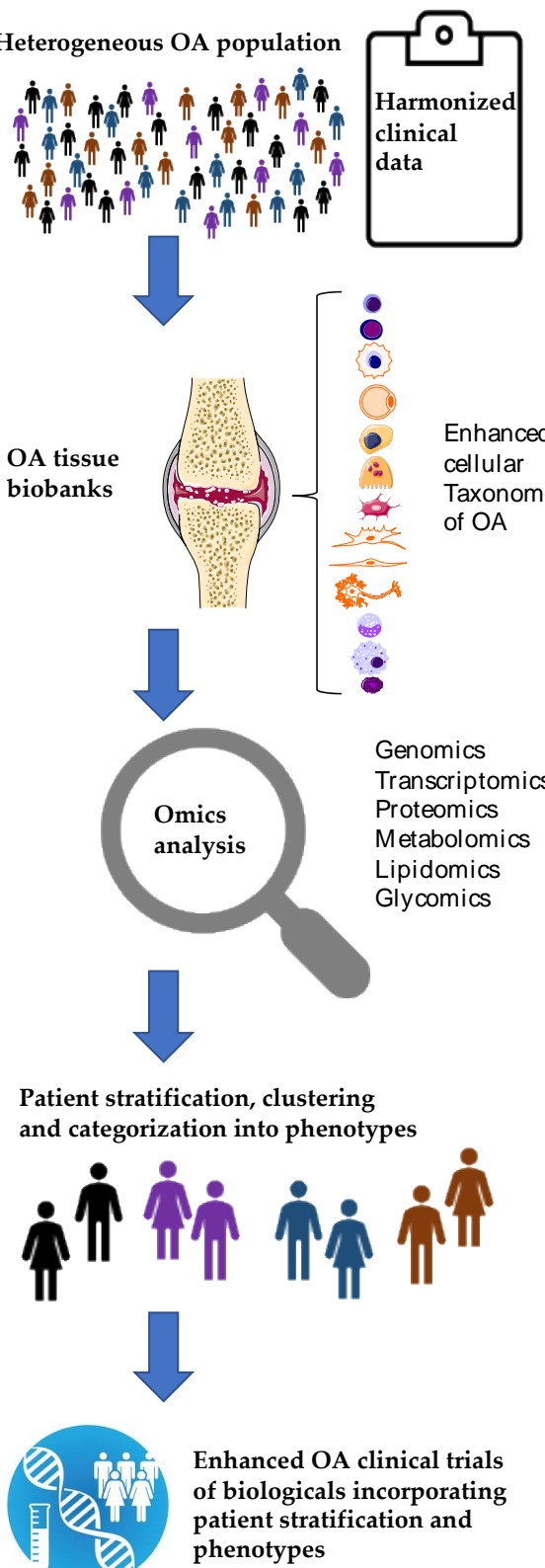

**Figure 4.** The proposed strategy for the phenotyping and stratification of a heterogeneous population of OA patients in future clinical trials. Patient stratification into different phenotypic categories further refine the initial clinical classification criteria and allow more effective targeting of biological drugs, including mAbs coming from neuroscience and cardiovascular research.

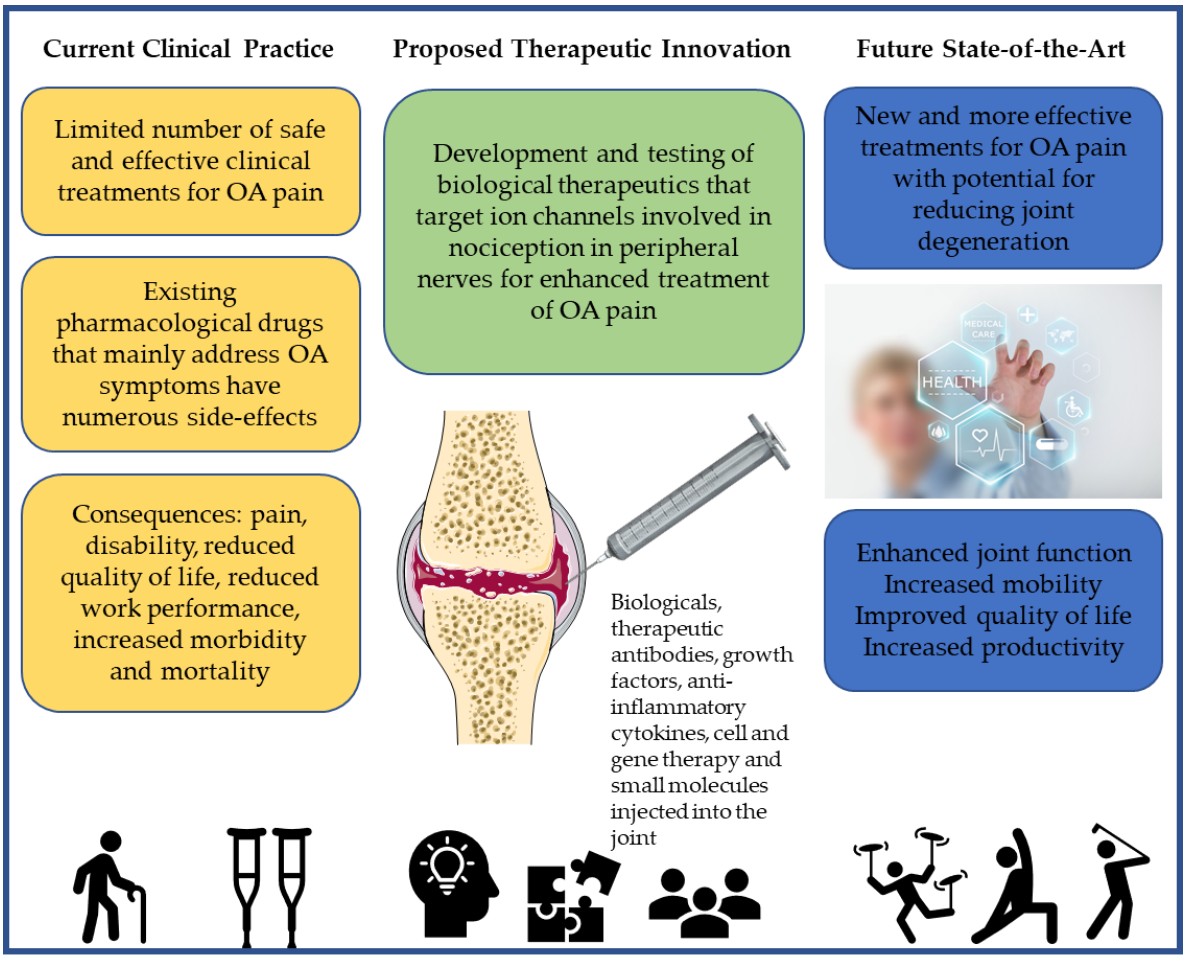

**Figure 5.** Current clinical practice and proposed intra-articular therapeutic innovations that will shape the future state-of-the-art in addressing OA symptoms and pain.

## 9. Conclusions

The aim of this review article was to highlight the roles of neuroscience and neuroimmunology in the development of new therapeutics for OA. Neuroscience can potentially offer new insights and solutions for OA pain. More research is needed to understand the role of the central and peripheral nervous systems in OA. We need to learn more about the physiology and pathophysiology of the peripheral innervation in OA and how neuro-inflammation of peripheral nerves can be selectively targeted for more successful outcomes in OA drug development. The intersection between the fields of neuroscience, immunology, orthopaedics and rheumatology can offer new solutions for OA pain and lead to the development of a new class of symptom and disease-modifying drugs that can target the key molecular players involved in the progression of pain (i.e., NGF) and related immunometabolic mechanisms, including the mechanistic target of rapamycin (mTOR) [24] that can provide the metabolic fuels needed to sustain and perpetuate the inflammatory pathways that eventually lead to pain in OA [131]. However, it is not sufficient to target pain as an isolated entity. It is also important to target the pathways that lead to its development, which is why research on new anti-inflammatory drugs and senolytic compounds targeting chondrosenescence mechanisms [132,133] needs to continue in conjunction with research on new analgesics that specifically target OA pain. Neuroscience and neuroimmunology are likely to impact more profoundly on orthopaedics and rheumatology and provide new therapies for OA symptoms and pain.

**Funding:** A.M. has received funding from the following sources: The European Commission Framework 7 program (EU FP7; HEALTH.2012.2.4.5-2, project number 305815; Novel Diagnostics and Biomarkers for Early Identification of Chronic Inflammatory Joint Diseases). The Innovative Medicines Initiative Joint Undertaking under grant agreement No. 115770, resources of which are composed of financial contribution from the European Union's Seventh Framework program (FP7/2007-2013) and EFPIA companies' in-kind contribution. A.M. also wishes to acknowledge funding from the European Commission through a Marie Curie Intra-European Fellowship for Career Development grant (project number 625746; acronym: CHONDRION; FP7-PEOPLE-2013-IEF). A.M. also wishes to acknowledge financial support from the European Structural and Social Funds (ES Struktūrinės Paramos) through the Research Council of Lithuania (Lietuvos Mokslo Taryba) according to the activity "Improvement of researchers" qualification by implementing world-class R&D projects' of Measure No. 09.3.3-LMT-K-712 (grant application code: 09.3.3-LMT-K-712-01-0157, agreement No. DOTSUT-215) and the and the new funding program: Attracting Foreign Researchers for Research Implementation (2018–2022).

**Institutional Review Board Statement:** Not applicable.

**Informed Consent Statement:** Not applicable.

**Data Availability Statement:** Not applicable.

**Acknowledgments:** The author wishes would like to acknowledge members of his research team in Vilnius, Lithuania, and collaborators in Oulu, Finland and Utrecht, The Netherlands for their support and encouragement.

**Conflicts of Interest:** The author declares no conflict of interest. The funding bodies that supported this work had no role to play in the decision to submit this manuscript.

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
