# Peer review of "Neuroscience and Neuroimmunology Solutions for Osteoarthritis Pain: Biological Drugs, Growth Factors, Peptides and Monoclonal Antibodies Targeting Peripheral Nerves"

_neurosci, doi:10.3390/neurosci2010003_

Round 1

Reviewer 1 Report

This communication summarizes some of the treatments for osteoarthritis and for pain and suggests areas of investigation that might further the search for disease modifying osteoarthritis drugs. In general this is a worthy topic. However the way this information was presented was somewhat disorganized and disjointed. Some of the statements are unclear in their meaning. The conclusion section brings in a host of new concepts that had not been previously discussed. Some, such as the information about obesity, inflammation, and diabetes starting line 232 could have been introduced in the section 2. Osteoarthritis. The paragraph starting line 242 brings in a completely new idea regarding nanobodies and bispecific antibodies. It is not clear to my whey these were introduced in the Conclusions.  IL-1 inhibitors are introduced as a treatment for OA in the first paragraph of Conclusions but should have been introduced in the Section 4. Emerging biological treatments for OA. Section 5 Monocloncal Antibodies Targeting Ion Channels would be better placed after section 7. Editorial changes are suggested as follows:

line 43 -  delete "on"

line 51 - change address to addresses

line 66 - delete "sex and"

line 86 - change "date producing" to "date have produced"

Lines 86-91 - I am not sure what this sentence is trying to say

line 100 - delete "produce"

line 102 - change "readers to" to "readers are directed to"

line 112 - insert "are" before "injected" and delete "orally and" as mAbs are never given orally

line 120 - delete "by"

line 122 - insert "the" before "complete"

Figure 3 - suggest changing X to No for clarity

Figure 5 - middle yellow box - insert "that" before "mainly"

line 229 - change "be" to "the"

Author Response

Response to Reviewer 1

This communication summarizes some of the treatments for osteoarthritis and for pain and suggests areas of investigation that might further the search for disease modifying osteoarthritis drugs. In general this is a worthy topic. However the way this information was presented was somewhat disorganized and disjointed. Some of the statements are unclear in their meaning. The conclusion section brings in a host of new concepts that had not been previously discussed. Some, such as the information about obesity, inflammation, and diabetes starting line 232 could have been introduced in the section 2. Osteoarthritis. The paragraph starting line 242 brings in a completely new idea regarding nanobodies and bispecific antibodies. It is not clear to my whey these were introduced in the Conclusions.  IL-1 inhibitors are introduced as a treatment for OA in the first paragraph of Conclusions but should have been introduced in the Section 4. Emerging biological treatments for OA. Section 5 Monoclonal Antibodies Targeting Ion Channels would be better placed after section 7. Editorial changes are suggested as follows:

Author’s Response: Thank you for taking the time to provide such detailed and helpful comments on my review article. I’m very grateful for the effort that you have put into your report and I hope that my responses below and the edits that I have made to the revised manuscript will meet with your approval. In your report above you make some specific suggestions for revisions. I agree with your suggestions and I have made the necessary modifications to the manuscript. I have expanded the idea of antibody therapy, focusing on nanobodies and bispecific antibodies. I also agree that some parts of the text were disorganised and disjointed and your comments have helped me improve the flow and readability of the paper and joining the arguments together more effectively.

line 43 -  delete "on"

Author’s Response: Thank you for pointing this out. This has been changed as you requested.

line 51 - change address to addresses

Author’s Response: Thank you for pointing this out. This has been changed as you requested.

line 66 - delete "sex and"

Author’s Response: Thank you for pointing this out. This has been changed as you requested.

line 86 - change "date producing" to "date have produced"

Author’s Response: Thank you for pointing this out. This has been changed as you requested.

Lines 86-91 - I am not sure what this sentence is trying to say

Author’s Response: I sincerely apologise for the ambiguity, which has been addressed.

line 100 - delete "produce"

Author’s Response: Thank you for pointing this out. This has been changed as you requested.

line 102 - change "readers to" to "readers are directed to"

Author’s Response: Thank you for pointing this out. This has been changed as you requested.

line 112 - insert "are" before "injected" and delete "orally and" as mAbs are never given orally

Author’s Response: Thank you for pointing this out. This has been changed as you requested.

line 120 - delete "by"

Author’s Response: Thank you for pointing this out. This has been changed as you requested.

line 122 - insert "the" before "complete"

Author’s Response: Thank you for pointing this out. This has been changed as you requested.

Figure 3 - suggest changing X to No for clarity

Author’s Response: Thank you for pointing this out. This has been changed as you requested.

Figure 5 - middle yellow box - insert "that" before "mainly"

Author’s Response: Thank you for pointing this out. This has been changed as you requested.

line 229 - change "be" to "the"

Author’s Response: Thank you for pointing this out. This has been changed as you requested.

Reviewer 2 Report

Well presented.  Could be a little shorter but since this is more of a review rather than original research paper.  

Author Response

Response to Reviewer 2

Well presented.  Could be a little shorter but since this is more of a review rather than original research paper.

Author’s Response: I sincerely wish to thank reviewer 2 for the positive and constructive comments about my review. I was pleased to learn that reviewer 2 thought that my paper was well presented. I apologise for the long length of the paper, because it is a review article rather than an original research paper. Additional edits have been made to the manuscript to address the comments of reviewer 1 and reviewer 3.

Reviewer 3 Report

Paper must be improved.

  1. It is not clear what is the real topic of this paper, please highlight it in Introcuction section (lines 30-42).
  2. In "Emerging Biological Treatments for OA" (lines 93-104) some biomechanics concepts need to be introduced. Just few lines. Please add these two very important references: Montemurro N et al. The Y-shaped trabecular bone structure in the odontoid process of the axis: a CT scan study in 54 healthy subjects and biomechanical considerations. J Neurosurg Spine. 2019 Feb 1:1-8. doi: 10.3171/2018.9.SPINE18396.    Krüger A et al. Height and volume restoration in osteoporotic vertebral compression fractures: a biomechanical comparison of standard balloon kyphoplasty versus Tektona® in a cadaveric fracture model. BMC Musculoskelet Disord. 2021 Jan 13;22(1):76. doi: 10.1186/s12891-020-03899-7. 
  3. In Discussion section must be highlight most recent innovations and state of art, please improved just 70-100 words more
  4. Conclusion is too long, try to reduce it. Maybe some sentences can be moved to the Discussion section

Author Response

Response to Reviewer 3

Paper must be improved.

Author’s Response: Thank you for your constructive comments. I have endeavoured to improve the manuscript along the lines suggested by reviewer 3 and the other two reviewers.

  1. It is not clear what is the real topic of this paper, please highlight it in Introduction section (lines 30-42).

Author’s Response: Thank you for your comment. The real topic of this paper is the importance of biological therapy for OA, especially therapeutic concepts that have emerged from the field of neuroscience. /I accept that this aspect did not come out strongly enough in the submitted version of the manuscript. The revised version makes it clear what the paper is all about and focuses on the topic of emerging biological therapies from the intersection between neuroscience, rheumatology and orthopaedics.

  1. In "Emerging Biological Treatments for OA" (lines 93-104) some biomechanics concepts need to be introduced. Just few lines. Please add these two very important references: Montemurro N et al. The Y-shaped trabecular bone structure in the odontoid process of the axis: a CT scan study in 54 healthy subjects and biomechanical considerations. J Neurosurg Spine. 2019 Feb 1:1-8. doi: 10.3171/2018.9.SPINE18396. Krüger A et al. Height and volume restoration in osteoporotic vertebral compression fractures: a biomechanical comparison of standard balloon kyphoplasty versus Tektona® in a cadaveric fracture model. BMC Musculoskelet Disord. 2021 Jan 13;22(1):76. doi: 10.1186/s12891-020-03899-7.

Author’s Response: Thank you for your suggestion. The author has included discussion of biomechanical concepts in the revised version of the manuscript. The two papers that you have suggested above are now cited in the revised version of the paper.

  1. In Discussion section must be highlight most recent innovations and state of art, please improved just 70-100 words more

Author’s Response: The author fully agrees with the reviewer. I have added another 100 words to the discussion section to highlight the most recent innovations and the current state of the art.

  1. Conclusion is too long, try to reduce it. Maybe some sentences can be moved to the Discussion section

Author’s Response: I agree, the conclusion is rather long. I have taken some of the sentences from the conclusion section and moved them to the discussion section as requested by the reviewer.

Author’s Response: I sincerely wish to thank reviewer 3 for the constructive comments that have helped to improve my review. I was pleased to learn that reviewer 3 thought that my paper was well presented. The revised introduction provide sufficient background and include all the relevant references. The conclusions in the revised manuscript of the manuscript are supported by the published literature. Additional edits have been made to the manuscript to address the comments of reviewer 1 and reviewer 2.

Round 2

Reviewer 1 Report

This version is much improved and the narrative flows in a logical and comprehensible way. 

Reviewer 3 Report

Good.